# Effect of Fiber Content and Alignment on the Mechanical Properties of 3D Printing Cementitious Composites

**DOI:** 10.3390/ma14092223

**Published:** 2021-04-26

**Authors:** Hao Zhang, Liming Zhu, Fan Zhang, Mijia Yang

**Affiliations:** 1Institute of Steel and Spatial Structures, College of Civil Engineering & Architecture, Henan University, Kaifeng 475004, China; zhciwei@163.com (H.Z.); zhangfan931005@163.com (F.Z.); 2Department of Civil and Environmental Engineering, North Dakota State University, Fargo, ND 58108-6050, USA

**Keywords:** fiber reinforced composites, mechanical properties, anisotropic behaviors, fiber alignment

## Abstract

This paper studies aligned glass fiber-reinforced composites for printing. To determine the influence of fiber content and alignment on the mechanical properties of this novel material, a large number of standard test specimens were prepared, which included samples fabricated by mold-casting, randomly dispersed fiber reinforced mixtures and aligned fiber cement composites containing 10 types of fiber volume ratios manufactured by nozzle sizes ranging of 24 and 10 mm (fiber length = 12 mm). Mechanical properties and failure modes of the specimens under compression and flexural tests were studied experimentally. The anisotropic behaviors of printed samples were analyzed by different loading directions. As a result, the compressive and flexural strength of printed samples showed obvious anisotropy. With the increase of fiber volume ratio, flexural strength of the fiber reinforced composite was elevated tremendously but its compression strength reduced slightly. Moreover, fiber alignment also had a significant influence on the mechanical properties of the fiber reinforced composite. The composite cement-based material with 1 vol.-% aligned fiber exhibited an excellent flexural strength of 9.38 MPa, which increased by 483% in comparison to that of the plain cement paste.

## 1. Introduction

3D printing cement technology, which is a cement moldless forming technology, can support free design of buildings based on layer by layer material superposition. As an emerging innovative technique which is extensively used in the construction industry, it has engaged wide attention and undergone tremendous developments. Developing reinforcing technologies is the core of 3D printing structural elements. Recent research projects have studied numerous new reinforcing techniques for 3D printing, including placement of penetrating bars through specific printed layers [1], the combination of conventional reinforcement [2] and inserting steel reinforcements into hollows reserved in the printed structure and filling with flowable concrete [3]. However, although these technologies provide effective improved bearing capacity for printing structures, they are contrary to the key idea of form-freedom and automation in 3D printing. Therefore, a common substitution to include high-performance synthetic fibers directly into the printed material is recommended [4].

Lawler et al. [5] concluded the mortar reinforced with different types of fibers delayed the formation of cracks, and consequently improved its pre-peak mechanical performance and strength. Le et al. [6] produced a high-performance fine aggregate concrete with 1% polypropylene microfibers and reached 107 MPa and 11 MPa for its 28 d compressive strength and flexural strength, respectively. Ma and Zhang [7] found that cement mortar containing chopped glass fiber and HPMC cellulose could increase the bonding ability of concrete, making it suitable for rapid prototyping of 3D printing buildings. The effects of different kinds of fibers on the mechanical properties of 3D printed cement-based materials were studied by Hambach and Volkmer [8]. It was found that cement-based materials mixed with 1 vol.-% of carbon fiber, or basalt fiber, or glass fiber showed a significant improvement of their flexural and compressive strengths. Ogura et al. [9] reported in 2018 that strain-hardening cement-based composites with 1% and 1.5% high-density polyethylene fiber contents could be adopted for digital construction and showed obvious strain-hardening characteristics. Nematollahi et al. [10] demonstrated that the addition of fiber increased the ductility and the flexural strength of 0.5 vol.-% polypropylene fiber mix material in the perpendicular direction more than that in the lateral direction. Panda et al. [11] investigated that the tensile and flexural properties of samples greatly were improved with the increase of glass fiber, but the compressive strength almost remained unchanged.

From the above literature review, the use of fibers can enhance the mechanical properties of the printed cement-based products, but the fact remains that randomly dispersed fibers in cement materials exhibit a limited improvement on flexural strength. During the 1990s, Aldea et al. had made attempts to extrude fiber-reinforced cement mortar by using a simple extrusion process in order to control the fiber alignment orientation and improve its flexural properties [12]. Fiber alignment has a potential effect on the mechanical properties of samples, but how to effectively control the fiber alignment is the key point. As noted by Akkaya et al. [13], 2 mm short fibers were more easily aligned during the extrusion process than 6 mm fibers and the ordered fiber alignment was beneficial to the performance of the extruded composite. Back in 2006, experiments showed that fibers could be aligned as a reinforcing agent for cement substrates by extrusion, and the toughness of aligned fiber composites was higher than the equal volume of randomly distributed fiber cement mortar [14]. Further research was conducted by StäHli et al. [15] through changing fiber distribution and orientation in fresh concrete to achieve a higher bending strength. Hambach et al. [16] researched the strengthened cement slurry with carbon fibers aligned by nozzles, which reached a high flexural strength of up to 100 MPa in 2016. Lee [17] developed a nozzle with blades called the B-nozzle and improved the fiber orientation and distribution in fiber-reinforced cement-based materials in 2019. The B-nozzle increased the mean fiber orientation coefficient by approximately 31~39% and the fiber distribution coefficient by 3~23%. By comparing aligned steel fiber cementitious composites and plain cement composite with randomly distributed steel fibers, fiber alignment contributed better performance in terms of uniaxial tensile strength and post-cracking toughness as demonstrated by Qing et al. [18]. Ma et al. [19] proposed a printed cementitious material with a good fiber alignment. The flexural strength of samples FZ was higher than that of the mold cast samples, samples FY and FX in the direction-based flexural tests.

The existing literature confirms that addition of fibers into mixed materials can increase their mechanical performance, and the orientation of fibers may have a great influence on the mechanical properties. However, for a comprehensive understanding of the influence of fiber content and alignment on mechanical properties of composites, the existing research is still insufficient. More specifically, how to control the fiber orientation lacks in-depth research.

This paper studies effects of fiber content and alignment on the flexural and compressive properties of samples fabricated by a printer. Samples with aligned fibers were fabricated through a 10 mm nozzle with different fiber volume ratios (0.2%, 0.4%, 0.6%, 0.8%, 1%, 1.2%, 1.4%, 1.6%, 1.8% and 2%). To summarize the influence of fiber distribution, the randomly dispersed fiber samples with same fiber content were fabricated by a 24 mm nozzle and mold-cast samples are prepared. Three-point bending and compressive tests were carried out on these specimens. The anisotropic behaviors were investigated for printed samples with 1% fiber volume content loaded in the X, Y and Z directions. The fiber distribution and mechanical behaviors are probed and analyzed by scanning the failure section of specimens with an electron microscope.

## 2. Experiment

### 2.1. Raw Materials and Mix Design

The mix materials used in this research are Portland cement, silica fume, water reducing agent, glass fiber, and water. Among them, the binder component is composed of the rapid hardening Portland cement P.O.42.5, obtained from Lanke Environmental Water Purification Material Factory (Zhengzhou, China) and a high-performance silica fume, purchased from Sichuan Langtian Company (Langtian, China). Their chemical compositions are presented in Table 1. The polycarboxylate superplasticizer was purchased from Suzhou Xingbang Cooperation (Suzhou, China), as a cement dispersing agent to improve dispersion of cement particles and reduce the slump loss.

In addition, the pre-chopped glass fibers had a length of 12 mm. The tensile strength as reported by the manufacturer was 2800 MPa and its elastic modulus was 71,000 MPa (Table 2). The glass fibers were obtained from Yongxing Glass Fiber Factory (Yongxing, China). Water used was ordinary tap water.

To study the influence of fiber content on materials in greater detail, the fiber contents added by volume in the experiments were 0.2%, 0.4%, 0.6%, 0.8%, 1%, 1.2%, 1.4%, 1.6%, 1.8%, and 2%, respectively. The mixed mortar containing each fiber content was extruded through 24 and 10 mm nozzles to fabricate samples. Conventionally mold-cast samples with 0 and 1 vol.-% fiber were prepared to draw comparisons. Except fibers, the other mix proportions by weight were listed in Table 3.

Five groups, G0, G1c, G1d, G1, G2, were selected as representatives to illustrate the mix design. G0 and G1c were plain cement paste and 1 vol.-% reinforced cement respectively, fabricated by casting. G1d was 1 vol.-% of glass fiber cementitious composites extruded by the 24 mm nozzle. G1 and G2 were cementitious extrudates with 1 vol.-% and 2 vol.-% fiber, extruded by the 10 mm nozzle. The effect of preparation technology on improving the strength of samples can be investigated by comparing the results of specimens prepared by casting and printing, with the same mix proportion (G1c, G1d and G1). While comparing the test results of G0 with G1c, the effect of the use of fibers on the mechanical properties of composite cement mortar could be analyzed. Moreover, the influence of fiber volume on mechanical properties of composite materials by comparing G1 and G2 could be summarized. Finally, to analyze the effect of the anisotropic nature of the printing process, G1 was loaded in three directions, including X, Y, Z.

### 2.2. Specimen Preparation

The mixing of cementitious mortar was performed with a Hobart mixer. Cement, silica fume and water reducing agent were mixed first in a dry state for 2 min at low speed. Water and glass fiber were then added and the mixture was stirred for 4 min until the fibers were uniformly dispersed. Finally, the fiber-reinforced cement paste was obtained.

For cement-based materials extrusion, a gantry-style 3D printer HC1009 from the CIIC Company (Tianjin, China) is adopted to manufacture specimens in this study. This setup consists of a gantry system, a pumping system, a print head containing a nozzle and a modified container, a build platform and is controlled through a LCD screen (Figure 1). A progressive cavity pump is used in this printer to provide pressure and deliver materials. When printing, the fresh mixture is filled into the storage bin and conveyed to the container by the pumping system. A small amount of mixture can be directly poured into the container through a feed port. The effective printing space size of this setup is 2 m in length, 1 m in width and 1 m in height. The circular nozzle is 10 mm in diameter with a vertical moving speed of 20 mm/s, and the size of the extruded filament was 10 mm in width and 5 mm in height. During the printing process, the horizontal printing speed for all specimens is set as 50 mm/s and interlayer interval time is designed as 60 s. Each layer height of 5 mm is adopted, so the machine is designed to move up in 5 mm intervals. The rectangular corners of the traditional container used by Hambach and Volkmer [8] often obstruct the material extrusion and cause the fibers to be stacked at the corner bottom. In this study, the improved container (Figure 1) with arc shape is adopted to avoid fiber accumulation and improve the flowability in the extrusion process.

Slabs with dimensions of 240 × 240 × 140 mm^3^ were extruded by the printer with 10 mm and 24 mm nozzles. After printing, the slabs were firstly covered with plastic wraps to reduce moisture evaporation and stored for 24 h in the standard curing room, then placed in a water bath for 6 days. After water bath, slabs were transferred to the standard curing room for another 28 days. Finally, they were saw-cut and polished into the specimens with designed dimensions. For casting, the specimens were fabricated into 40 × 40 × 160 mm^3^ and 50 × 50 × 50 mm^3^ standard molds and cured in the same way for reference.

### 2.3. Flowability

The jumping table test was conducted to determine the flowability of the fresh fiber-reinforced cement mortar, in accordance with GB/T 2419–2005 [20]. The fresh cement-based composite was filled into the cone mold (dimensions 100 mm base diameter, 70 mm top diameter and 150 mm height) according to the standard. After the cone was removed, the table was lifted-and-dropped 25 times in 25 ± 1 s, allowing cementitious material to spread on the plate. Then, the diameters of spread cementitious mixture in two perpendicular directions were measured by calipers and the average value was calculated to show the flowability of the fresh fiber reinforced cement mortar.

### 2.4. Flexural Strength

Three-point bending test on both casted and printed samples was carried out according to NEN-EN 196-1 [21]. The printed samples with a size of 40 × 40 × 160 mm^3^ were sawed from the printed slabs and tested along with casted samples of same size. To better determine the direction of applied load, a specific coordinate system was defined. The *X* axis was defined as fiber alignment direction, that was, all fibers were parallel to *X* axis. The *Y* axis was set as nozzle movement direction, which was perpendicular to *X* axis in the build platform. The *Z* axis was layers stacking direction, perpendicular to the build platform. During the flexural and compressive tests, printed cubic and prism samples were loaded from three directions: *X*, *Y* and *Z* as sketched in Figure 2. The printed samples loaded in the *X*, *Y* and *Z* directions were named F*x*, F*y* and F*z*, respectively. The testing machine adopted was a Micro servo hydraulic universal testing machine with a maximum load capacity of 3000 kN. All samples were loaded from 0 kN and the loading speed was set as 1 mm/min until failure. The maximum force *F* was recorded and the flexural strength fs could be determined through the following expression:(1)$$f_{s}=\frac{3}{2}\frac{Fl}{bh^{2}}$$
where *l* is the distance between the supports, here equal to 100 mm, *b* represents the specimen’s width, and *h* represents the specimen’s height.

### 2.5. Compressive Strength

All specimens (both mold-cast and printed) were employed to obtain the compressive strength, based on ASTM C109 [22]. Like in the flexural test, 50 mm cube samples were extracted from the 240 × 240 × 140 mm^3^ slabs. Three samples were tested for each testing series (Figure 2). The testing machine adopted was the same as that of the bending test. In order to ensure that the top and bottom surfaces of samples bear the pressure uniformly and eliminate the eccentric effect, the prepared samples were scraped flat by using a diamond blade before testing and placed in the center of the lower piston [23]. The speed of the loading piston was set as 1 mm/min to obtain the maximum force *F*, and then the compressive strength *f_c_* could be calculated as follows:(2)$$f_{c}=\frac{F}{b^{2}}$$
where *b* is the cubic specimen’s width and *f_c_* is the compressive strength.

### 2.6. Scanning Electron Microscopy (SEM)

The sample fragments obtained after the three-point bending test were chopped into specimens of 1.5 mm thickness with the same size. These specimens were then wet polished and glued by epoxy resin to microscope slides. In order to increase the conductivity of surface and reduce the electrical charge, the specimens were coated using a gold sputter coater and put into the machine to make them dry. Finally, the processed specimens were observed by scanning electron microscopy (FEI, Hillsboro, USA) in a vacuum environment to characterize the microstructure of the composite. Ten specimens were observed in each group. Attention was given to the direction of fiber alignment and fiber distribution of the fracture surface.

## 3. Results and Discussions

### 3.1. Fiber Micro-Reinforcement Mechanical Behavior

In this study, a revised container is adopted to improve the fluency of extrusion process, thereby acquiring a better fiber alignment than the traditional container. To verify this hypothesis, thin sections with the same size and thickness are prepared and observed by the scanning electron microscope (Figure 3). It is clearly seen that a small amount of fibers inside the samples fabricated by the traditional container presents a two-dimensional distribution in accordance with the angle between red lines. In contrast, the fiber distribution of samples manufactured by the revised container appears uniform and fibers are almost oriented along a single direction (Figure 3a,b). Thus, the revised container serves to get a good fiber alignment.

Figure 3c shows the fiber distribution of G1d extruded by the 24 mm nozzle. The fibers of the fracture surface are interlaced and the angle between fiber and the extrusion direction is more than 90°, which displays a random fiber orientation. For samples fabricated by mold-casting, the distribution of G1c is similar to G1d. In t contrast, the 10 mm nozzle-extruded fiber cement mortar at both 1 vol.-% and 2 vol.-% exhibits a good fiber alignment as shown in Figure 3d,e. The fibers are nearly aligned with the print direction and the density of aligned fibers in G2 is much higher in comparison to G1. It can be considered that fibers are fully aligned along the print direction when the angle between fiber and print direction within ±20° according to the proposed convention [19]. Therefore, it is proved that fibers within the mixed mortar will be aligned along the stress direction through high compression and high shear in the case of the fiber length being larger than the nozzle diameter. 

Additionally, Figure 3f is a G1 failure section under the three-point bending test, demonstrating the fiber reinforcement mechanism. During the loading process some fibers fracture, and a small amount of fibers are pulled out from the cement composite matrix. The process delays the propagation of cracks and restricts the deformation by consuming a huge amount of energy. Consequently, the mechanical properties of specimens are enhanced.

### 3.2. Effect of Fiber Volume Fraction on Flowability of Cementitious Material

The results of the jump table tests were obtained and listed in Table 4. They indicate that the flowability of mixed mortar is gradually reduced when the fiber content increases. Mixed mortars with 0.2~0.8 vol.-% fiber have higher flow spread diameters, and are squeezed out smoothly. For 1~1.4% fiber volume content, the composite paste can be extruded and shows good working performance. However, when the composite paste over 1.6 vol.-% fiber is used, the nozzle becomes clogged sometimes during the printing process, thus affecting the quality of specimens. Furthermore, if the fiber volume ratio reaches 2%, flowability of the cement composite materials is greatly reduced, which can easily block the nozzle.

### 3.3. Effect of Fiber Volume Fraction on Mechanical Properties of Testing Samples

In order to determine the effect of fiber volume fraction on the mechanical properties of the manufactured samples, the average compressive strengths of Fx and their bending strength-deflection curves of Fz were obtained (Figure 4).

Figure 4a shows the flexural strength versus mid-span deflection of specimens loaded in the Z direction. It can be found that fiber reinforced cementitious materials exhibit obvious deflection hardening behaviors until failure. The flexural strength of plain cement paste G0 is weaker than others, only up to 1.61 MPa. Average strength reductions of 69.5% and 69.7% compared to G1d and G1c are found. In addition, G0 shows a brittle failure as observed by the sudden decrease of load to zero when the specimen is cracked. Therefore, the addition of fiber improves the load carrying capacity of the cementitious composite after the first cracking and prevents samples from being completely destroyed. G1 and G2 are samples containing the aligned fiber with 1% and 2% fiber volume, which have an excellent flexural strength of 9.38 MPa and 12.05 MPa, respectively. The flexural strength of G2 is improved by 28% compared to that of G1. It can be seen from this that fiber content has a great influence on the flexural strength of specimens. When the fiber volume content increases from 0 to 1%, the bending strength of samples with aligned fiber is increased by 483% and that of randomly dispersed fiber samples is enhanced by 226%. If the fiber volume content increases from 1% to 2%, the bending strength of samples with aligned fiber is improved by 28% and the bending strength of randomly dispersed fiber samples is boosted by 88%.

In addition to increasing material strength, toughness enhancement is noticed with addition of aligned fiber. The flexural toughness of printed samples F*z* with 0~1 vol.-% aligned fiber is calculated from the area under the flexural strength versus mid-span deflection showed in Figure 4a up to mid-span deflection of 2.25 mm (Figure 4b). A steady increase of flexural toughness is found as more fibers are used. It is obvious that G0 shows poor toughness, while G1 exhibits higher toughness of 9.56 J/mm^2^ than others. Addition of fiber has a positive effect on fracture behavior, which is similar to the results obtained by Nematollahi et al. [10].

Figure 4c presents the compression strength of tested samples in the X direction which ranges from 24.63 to 16.3 MPa. The compressive strengths of samples are reduced gradually as the fiber content increases. Samples G0 prepared by conventional mold-casting show the excellent compressive strengths at 24.63 MPa. For samples fabricated by the 10 mm nozzle, the compressive strengths of G1 and G2 are reduced to 17.34 and 16.3 MPa, respectively. One of the possible explanations is addition of fiber introduces additional voids resulting in weak bonding interfaces between fiber and cement. As more fiber employed, the number of weak bonding interfaces also increases, thus weakening the structural integrity of samples and decreasing the compressive strength [24].

### 3.4. Effect of Fiber Alignment on the Mechanical Properties of Test Samples

Figure 5a shows the measured results of samples fabricated by different preparation methods under 3-point bending tests of F*z*. Samples fabricated by mold-casting exhibit a poor flexural strength of about 1.61 MPa when fiber content is 0, and 5.31 MPa if 1 vol.-% fiber is added. When printing samples with the same mix proportion, the bending strength is improved notably. 1 vol.-% aligned fiber samples G1 have a remarkable flexural strength of up to 9.38 MPa which is 5.83 times as much as that of G0. Compared to that of G1d and G1c, the flexural strengths are increased by 78% and 76.6%. It can thus be concluded that the bending performance of the samples with aligned fibers is better than that with randomly distributed fibers. This can be attributed to use of aligned fiber that reduces stresses in cement due to parallel fibers along the tensile direction and delays the propagation of cracks. If 2% fiber volume content is employed, the flexural strength of G2 reaches 12.05 MPa, elevated by 28.5% compared to G1 and enhanced by 7% than that of G1.6. It indicates that the strength gap decreases with the increase of fiber volume.

Figure 5b reveals the F*x* compressive test results of specimens manufactured by different preparation methods. It shows that G0 gets maximum strength of 24.63 MPa and the compressive strengths of samples manufactured by a 24 mm nozzle are greater than that of the samples extruded by a 10 mm nozzle under the same fiber content condition. The compressive strengths of G1, G1d and G1c measures 17.34 MPa, 20.32 MPa and 21.56 MPa. Compared with G0, they are reduced by 29.6%, 17.5% and 12.5%, respectively. The compressive strength of G2d is 1.3% higher than that of G2, measured at 16.3 MPa. This confirms that the extrusion process forms an interspace between adjacent layers and reduces the strength of the extruded fiber reinforced composite. In comparison to randomly dispersed fiber, fiber aligned a single direction has limited effect on the weak interface between layers. Moreover, if the aligned fiber volume ratio increases from 1.6% to 2%, the compressive strength of the extruded fiber reinforced composite does not change obviously anymore.

The variation of flexural strength measurement with respect to different loading directions for G1 is presented in Figure 5c. It is found that the flexural strength of printed samples shows an anisotropic behavior. The mean flexural strengths of G1 reach 4.26 MPa, 7.52 MPa and 9.38 MPa in *X*, *Y* and *Z* directions, while that of mold-cast specimens G1c is up to 5.31 MPa. Compared with G1c, the flexural strengths of G1 in the F*z* and F*y* directions are increased by 76.6% and 41.6%, respectively, and decreased by 19.8% in the F*x* direction. Similar phenomena were observed from Le et al. [6]. The flexural strength is determined by the center bottom of the beam specimen where the maximum tensile stress occurs and G1 tested in the F*z* direction is at the bottom of the printed slab where is probably well compacted in comparison to the F*y* direction. However, G1 of F*x* direction exhibits the lowest flexural strength. This is because that tensile stress acts perpendicular to the weak bond between the printed filaments. Figure 5d demonstrates that the compressive strength of G1 varies depending on the loading direction. It can be found that the highest strength of 21.56 MPa is achieved by G1c which is higher than that of G1 in any testing directions. This is explained by the addition of fiber increasing entrapped air inside cement and the lack of vibration during the printing process [25]. The average compressive strength of G1 F*x* is 16.6% and 42.7% higher than that of F*y* measured 14.87 MPa and F*z* of 12.15 MPa, respectively. This can be due to fiber parallel to the direction of the force and resisting the pressure to some extent when loading in F*x* the direction, and thus a higher strength than F*y* and F*z*.

Fibers exposed on the fracture surface are shown in Figure 6. As shown in Figure 6a, the fracture surface of G0 is flat, where no fine cracks occur. For the case of G1d (Figure 6b), the fracture section is uneven and rough, and the fibers on the fracture surface are staggered in a non-uniform manner. Figure 6c,d show the facture surface of G1 and G2, respectively. It can be seen that the fibers are clearly aligned along the extrusion direction at the cross section. The fibers on the G1 fracture surface are relatively sparse and evenly distributed, and the end of the fiber is tightly bonded to the cement. In comparison, the fibers on the G2 fracture surface are closely bonded to the cement side and make the cross-section rougher. This phenomenon, most of fibers in the mixed mortar formed by extrusion will be oriented in the stress direction when the fiber content is small, is consistent with the study by Feng et al. [26].

### 3.5. Failure Mode

The typical failure modes of the extruded fiber reinforced composite are shown in Figure 7 for the 3-point bending test of F*z*. The plain cement extrudates exhibit an obvious brittle failure mode (Figure 7a). In the loading process, microcracks occur in the midspan section of G0. When the ultimate stress is reached, the specimens are broken directly along the underside of the round steel and there is no obvious warning sign before the failure. Compared with G0, G1d shows a favorable integrity (Figure 7b). In the failure process, several curving cracks have appeared in the span of the sample before it is split from the middle. During the fiber fracture and pullout processes, a great number of energies are consumed and the fracture development is delayed to some extent [27,28,29,30,31]. Samples with aligned fibers have similar failure patterns as shown in Figure 7c,d. In the initial loading process, there are no obvious deformations and small cracks generated on the surface. With an increase in the load value, the bottom of samples near the support shows some deflection cracks which slowly expands to the top surface of the mid span. After the samples are broken, no huge blocks of composite cement falling off are observed. It can be concluded that the use of aligned fiber improves the carrying capacity of samples and changes the fracture patterns of samples from brittle fracture to ductile fracture.

Figure 8 shows the failure patterns of the cube specimens under compressive tests of F*x*. As can be seen from Figure 8a, G0 shows the shear tensile failure under the hooping force, resulting in cone-shaped fracture lines. Once the peak load is reached, sectional cracks run through samples and the strength drops sharply. However, after adding dispersed fibers into the cement material, there are microscopic cracks existing inside and on the surface of the specimens. As the load continues to increase, the large compressive stress of the specimens makes the cracks expand constantly. The failure mode is not changed obviously compared to G0, although the ductility of samples is relatively improved (Figure 8b).

Figure 8c,d show the failure patterns of G1 and G2 which are different from the G1d where the fibers are randomly dispersed. A small number of irregular and subtle cracks occur on the surface under the early load. With the load increasing, the cracks on the specimen surface show an obvious trend extending along the vertical direction. That is probably because the fibers parallel to the pressure form an evenly distributed outline of short columns. The interfaces between adjacent columns do not have the effective cohesiveness, which is prone to cause the separation of the layers to shape as vertical lines.

## 4. Conclusions

In this paper, composite cement samples with different fiber volume ratio are prepared by a printer through 10 and 24 mm nozzles. These samples are divided into 20 groups to test their flexural and compressive strengths and compare them with samples fabricated by mold-casting. By comparing their mechanical properties and failure patterns, the effects of fiber content and fiber alignment are analyzed. The anisotropic mechanical behaviors of printed samples are investigated under different loading directions. The conclusions can be summarized as folllows:(1).With the increase of fiber volume ratio, the flexural strengths of samples will increase dramatically while the compressive strengths will be decreased slightly. For example, compared with G0, the flexural strength of G1 achieves 9.38 MPa increased by 483% and G1d measures 5.28 MPa enhanced by 228%, while the compressive strength of G1 and G1d are down to 17.34 MPa and 20.32 MPa decreased by 29.6% and 17.5%.(2).Fiber alignment has an important impact and is a good method to significantly improve the flexural properties of specimens. Compared with G1d, the flexural performance of G1 has a relative enhancement of 78% and that of G2 is relatively enhanced by 128%.(3).The application of the revised container by changing the corners from rectangle to arc is meaningful to improve the smoothness in the extrusion process. More importantly, the revised container has a better fiber alignment than the traditional container from SEM images.(4).The nozzle diameter plays an important role in fiber alignment. When the nozzle diameter is smaller than that of the fiber, fiber in the mixed cement will be aligned along the stress direction. If the nozzle diameter is much larger than the fiber length, the fibers of the composite material are randomly distributed.(5).The printed samples show an obvious anisotropic behavior. The flexural strengths of G1 of F*z* and F*y* are increased by 76.6% and 41.6%, respectively, while they are decreased by 19.8% in the *X* direction compared with G1c. For the compression test, the average strengths of G1 of F*x*, F*y* and F*z* are 19.6%, 31% and 43.6% lower than that of G1c.

## Figures and Tables

**Figure 1 materials-14-02223-f001:**
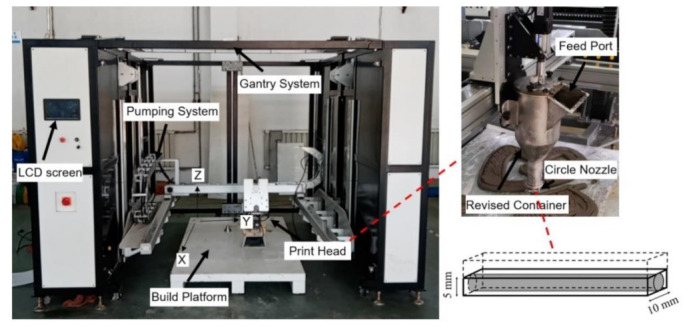
The gantry-style 3D printer HC1009 and the size of the extruded filament.

**Figure 2 materials-14-02223-f002:**
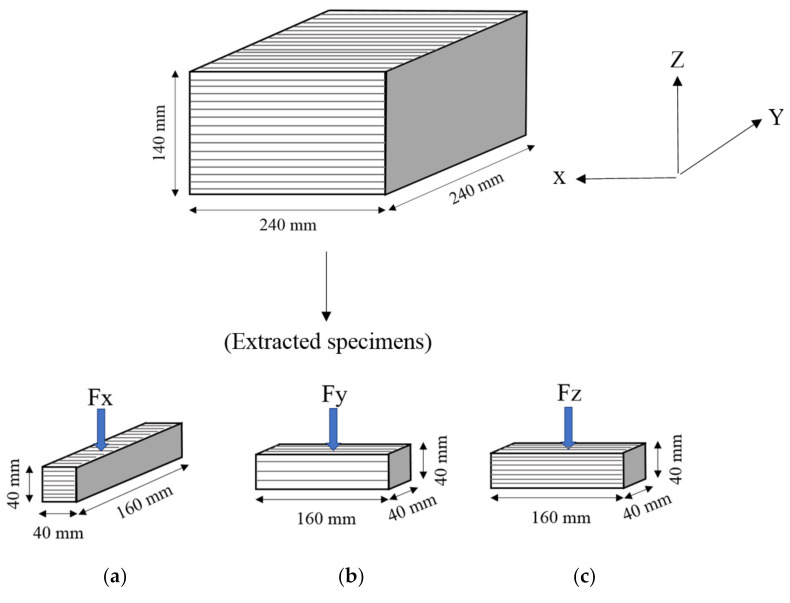
Loading directions for 3-point bending tests (**a**–**c**) and compression tests (**d**–**f**).

**Figure 3 materials-14-02223-f003:**
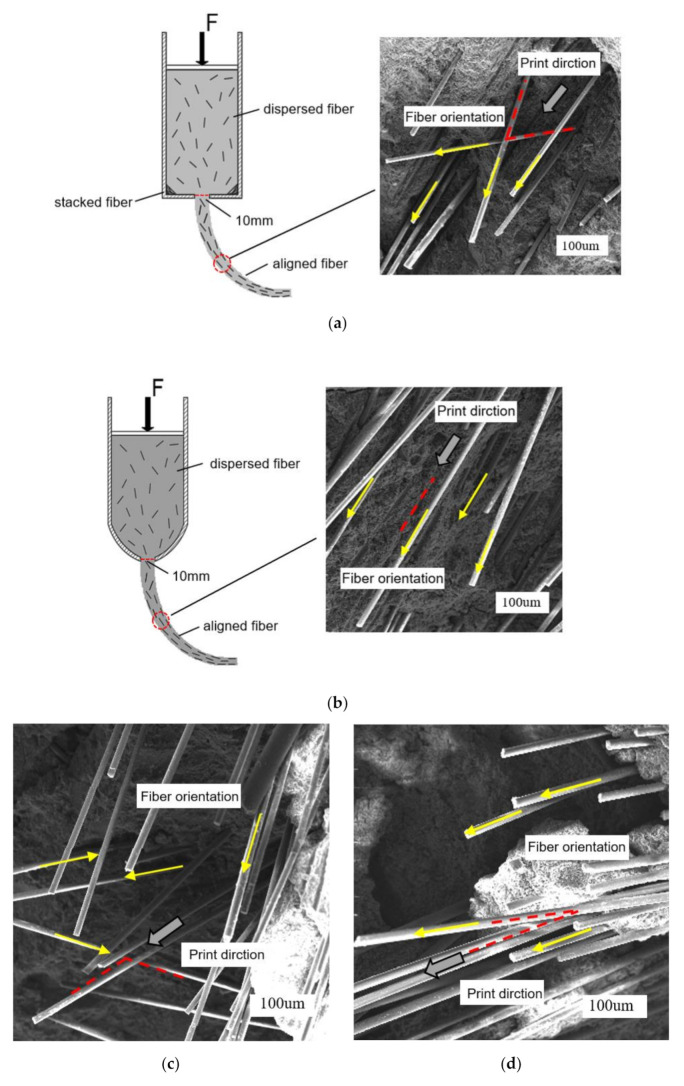
SEM images of fiber orientation under (**a**) traditional container, (**b**) revised container and the fracture surface of (**c**) G1d, (**d**) G1, (**e**) G2, (**f**) broken fiber.

**Figure 4 materials-14-02223-f004:**
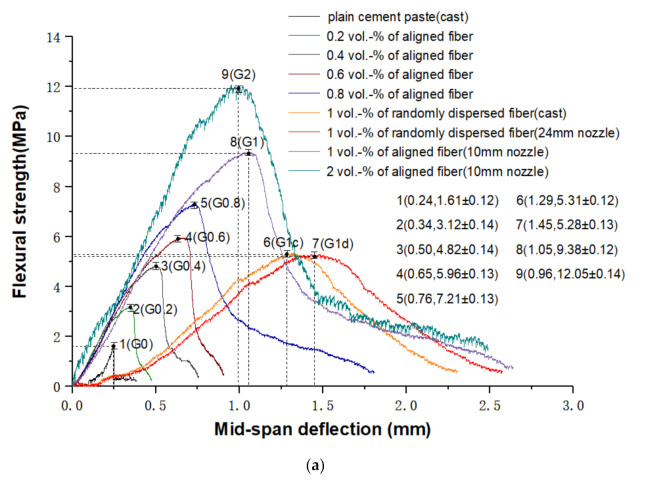
(**a**) The flexural strength versus mid-span deflection curves for samples loaded in the Z direction. (**b**) Flexural toughness of printed samples F*z* with 0 to 1 vol.-% aligned fiber. (**c**) Compressive strength of samples with aligned fiber in the F*x* direction.

**Figure 5 materials-14-02223-f005:**
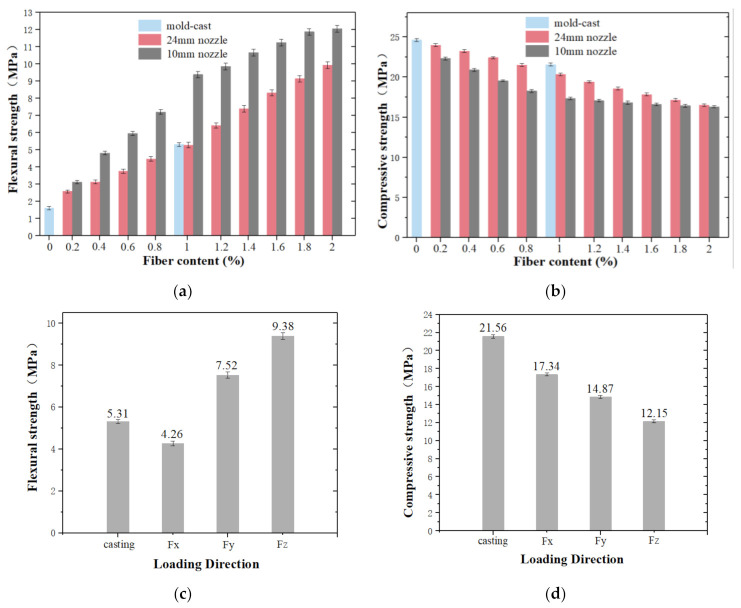
(**a**) Flexural strength of samples fabricated by different preparation methods under 3-point bending tests of F*z*. (**b**) Compressive strength of samples fabricated by different preparation methods under compression tests of F*x*. (**c**) Flexural and (**d**) compressive strengths of printed specimens in the different loading directions.

**Figure 6 materials-14-02223-f006:**
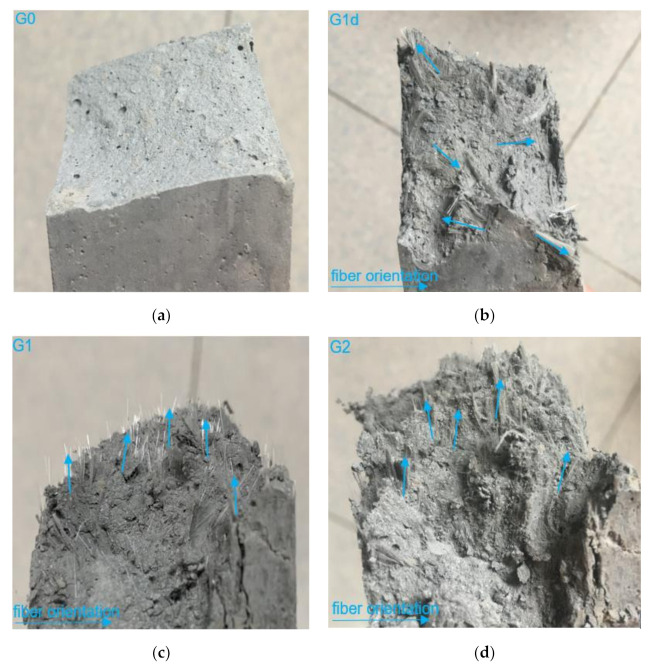
Failure section photographs of (**a**) G0, (**b**) G1d, (**c**) G1, (**d**) G2.

**Figure 7 materials-14-02223-f007:**
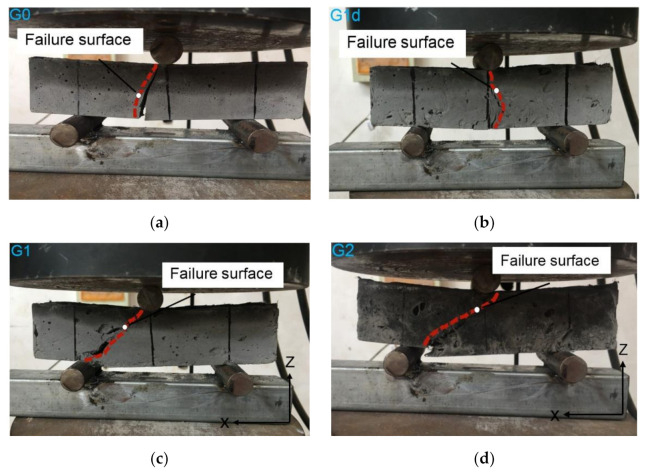
Failure patterns of (**a**) G0, (**b**) G1d, (**c**) G1 and (**d**) G2.

**Figure 8 materials-14-02223-f008:**
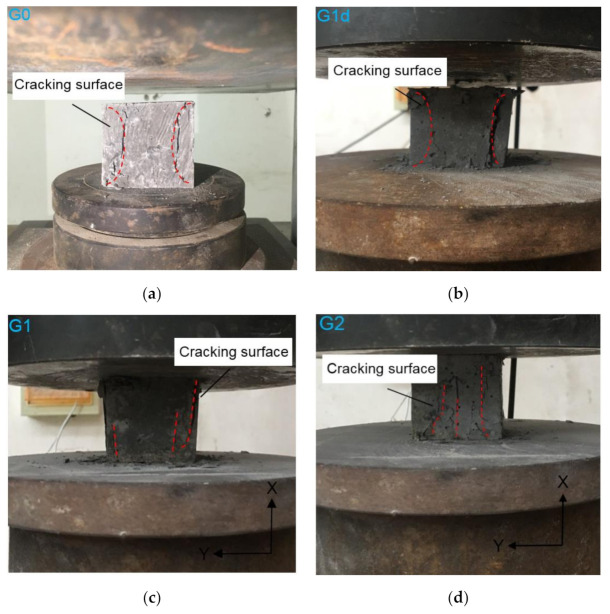
(**a**) Failure patterns in compressive tests of F*x* for (**a**) G0, (**b**) G1d, (**c**) G1 and (**d**) G2.

**Table 1 materials-14-02223-t001:** The chemical compositions of cement P.O.42.5 and silica fume.

Component	Average Percentage by Weight (%)	
	Cement P.O.42.5	Silica Fume
SiO_2_	19.4–21.5	90–97
Al_2_O_3_	4.1–4.9	
Fe2O3	2.7–2.9	
CaO	61.9–64.1	
MgO	1.1–1.2	
SO_3_	3.0–3.2	
K_2_O	0.6–0.7	
Na_2_O	0.2	
Cl	0.02–0.05	
Loss of ignition	2.3–4.1	<3.0

**Table 2 materials-14-02223-t002:** Properties of the glass fibers.

Density (g/cm^3^)	Elastic Modulus (MPa)	Tensile Strength (MPa)	Length (mm)	Diameter (µm)	Elongation (%)	Color
2.54	71,000	2800	12	9	2.6	White

**Table 3 materials-14-02223-t003:** Mix proportions.

**Mix ID**		Weight Fraction			Fiber (by Volume)	Preparation Method
	Cement	Water Reducer Agent	Silica Fume	Water		
G0	0.53	0.18	0.27	0.02	0%	Cast
G1c	0.53	0.18	0.27	0.02	1%	Cast
G0.2	0.53	0.18	0.27	0.02	0.2%	10 mm nozzle
G0.4	0.53	0.18	0.27	0.02	0.4%	10 mm nozzle
G0.6	0.53	0.18	0.27	0.02	0.6%	10 mm nozzle
G0.8	0.53	0.18	0.27	0.02	0.8%	10 mm nozzle
G1	0.53	0.18	0.27	0.02	1%	10 mm nozzle
G1.2	0.53	0.18	0.27	0.02	1.2%	10 mm nozzle
G1.4	0.53	0.18	0.27	0.02	1.4%	10 mm nozzle
G1.6	0.53	0.18	0.27	0.02	1.6%	10 mm nozzle
G1.8	0.53	0.18	0.27	0.02	1.8%	10 mm nozzle
G2	0.53	0.18	0.27	0.02	2%	10 mm nozzle
G0.2d	0.53	0.18	0.27	0.02	0.2%	24 mm nozzle
G0.4d	0.53	0.18	0.27	0.02	0.4%	24 mm nozzle
G0.6d	0.53	0.18	0.27	0.02	0.6%	24 mm nozzle
G0.8d	0.53	0.18	0.27	0.02	0.8%	24 mm nozzle
G1d	0.53	0.18	0.27	0.02	1%	24 mm nozzle
G1.2d	0.53	0.18	0.27	0.02	1.2%	24 mm nozzle
G1.4d	0.53	0.18	0.27	0.02	1.4%	24 mm nozzle
G1.6d	0.53	0.18	0.27	0.02	1.6%	24 mm nozzle
G1.8d	0.53	0.18	0.27	0.02	1.8%	24 mm nozzle
G2d	0.53	0.18	0.27	0.02	2%	24 mm nozzle

**Table 4 materials-14-02223-t004:** Spread out diameters of fiber reinforced cement mortar with different fiber content volume.

**Fiber Content (%)**	0.2	0.4	0.6	0.8	1	1.2	1.4	1.6	1.8	2
**Spreading Diameter (mm)**	241	225	211	197	182	173	158	143	129	112

## Data Availability

Some or all data, models, or code that support the findings of this study are available from the corresponding author upon reasonable request.

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
