# Peer review of "Effect of Fiber Content and Alignment on the Mechanical Properties of 3D Printing Cementitious Composites"

_materials, 2021, doi:10.3390/ma14092223_

Round 1
Reviewer 1 Report
The authors presented an experimental study for the effect of the fiber content and alignment on the flexural and compression properties of printed cementitious composite reinforced with short glass fiber. The study is followed by fracture analysis under these tests. This study contributes in valorization of 3D printing cementitious composite and aids to develop this technology. However, certain ambiguities need to be removed and some corrections are required.

Reviewer 2 Report
Dear authors,
Thank you so much for this extensive research in the field of 3D printing of concrete composites with fibers.
I appreciate the references, but there is more information in the area.
I'm a little confused by the information from the abstract and the conclusion. The abstract shows an improvement of 648% in bending, but the conclusion shows different results. Is it a mistake or, am I bad understand?
I lack a clearly specified reason in the introduction and the possibilities of using the technology that you presented. This is of course related to the choice of concrete mix and the choice of fiber material.
In the beginning, you show that there are compressive and flexural strengths in the literature "reached 107 MPa and 11 MPa", but you do not achieve such strengths - what is the advantage of your concrete mixtures?
Do you take anisotropy behavior as an advantage or a disadvantage?
A few things about the format:
- at the end of the article you have twice "Conflicts of interest"
- References have different formats. I don't think they are okay according to the template of the journal.
- You have a gray bar below the pictures.
- it is necessary to place the pictures so that they are on one side.
- would help unify the font of graphs and size to the same value.
- I recommend language correction - the text is okay to read, but there are special sentences sometimes.
Thank you
Regards
Round 2
Reviewer 1 Report
The authors considered carefully the mentioned remarks and removed the reported ambiguities.
However, the definition of Fx, Fy and Fy still lead to some misunderstanding. Do they mean the load compenents according to x, y and z direction? Or they are used to refere to flexural test in those directions? That leeds to confusion in reading the results section (3) with figure 2.
Reviewer 2 Report
Dear authors,
Thank you very much. Your reactions and answers are fine.
I will recommend the article for publication.
Regards,
Author Response
Thanks very much for your kind work and consideration on publication of our paper. On behalf of my co-authors, we would like to express our great appreciation to you.